# Stenting in Brain Hemodynamic Injury of Carotid Origin Caused by Type A Aortic Dissection: Local Experience and Systematic Literature Review

**DOI:** 10.3390/jpm13010058

**Published:** 2022-12-27

**Authors:** Jean-François Aita, Thibault Agripnidis, Benoit Testud, Pierre-Antoine Barral, Alexis Jacquier, Anthony Reyre, Ammar Alnuaimi, Nadine Girard, Farouk Tradi, Paul Habert, Vlad Gariboldi, Frederic Collart, Axel Bartoli, Jean-François Hak

**Affiliations:** 1Department of Neuroradiology, APHM La Timone, 13005 Marseille, France; 2LIIE, Aix Marseille University, 13007 Marseille, France; 3CERIMED, Aix Marseille University, 13007 Marseille, France; 4Department of Medical Imaging, APHM La Timone, 13005 Marseille, France; 5Department of Cardiac Surgery, La Timone Hospital, La Timone Hospital 264, Rue Saint Pierre, 13005 Marseille, France

**Keywords:** aortic, dissection, brain, malperfusion, carotid, stenting, surgery

## Abstract

In this study, we report our local experience of type A aortic dissections in patients with cerebral malperfusion treated with carotid stenting before or after aortic surgery, and present a systematic literature review on these patients treated either with carotid stenting (CS) before or after aortic surgery (AS) or with aortic and carotid surgery alone (ACS). We report on patients treated in our center with carotid stenting for brain hemodynamic injury of carotid origin caused by type A dissection since 2018, and a systematic review was conducted in PubMed for articles published from 1990 to 2021. Out of 5307 articles, 19 articles could be included with a total of 80 patients analyzed: 9 from our center, 29 patients from case reports, and 51 patients from two retrospective cohorts. In total, 8 patients were treated by stenting first, 72 by surgery first, and 7 by stenting after surgery. The mean age; initial NIHSS score; time from symptom onset to treatment; post-treatment clinical improvement; post-treatment clinical worsening; mortality rate; follow-up duration; and follow-up mRS were, respectively, for each group (local cohort, CS before AS, ACS, CS after AS): 71.2 ± 5.3 yo, 65.5 ± 11.0 yo; 65.3 ± 13.1 yo, 68.7 ± 5.8 yo; 4 ± 8.4, 11.3 ± 8.5, 14.3 ± 8.0, 0; 11.8 ± 14.3 h, 21 ± 39.3 h, 13.6 ± 17.8 h, 13 ± 17.2 h; 56%, 71%, 86%, 57%; 11%, 28%, 0%, 14%; 25%, 12.3%, 14%, 33%; 5.25 ± 2.9 months, 54 months, 6.8 ± 3.8 months, 14 ± 14.4 months; 1 ± 1; 0.25 ± 0.5, 1.3 ± 0.8, 0.68 ± 0.6. Preoperative carotid stenting for hemodynamic cerebral malperfusion by true lumen compression appears to be feasible, and could be effective and safe, although there is still a lack of evidence due to the absence of comparative statistical analysis. The literature, albeit growing, is still limited, and prospective comparative studies are needed.

## 1. Introduction

Type A aortic dissection (AAD) is a diagnostic and therapeutic emergency, always needing an urgent surgical repair.

Left untreated, mortality is 1–2% per hour, 25% the first day, 50% in 48 h and 75% in two weeks. Incidence is estimated at 3–6 people per 100,000 people per year [1].

Complications are related to extension of the dissection; retrograde extension to the aortic ring can lead to acute aortic insufficiency and to an intra-pericardial effusion, with cardiac tamponade, accounting for 70% of acute mortality. False lumen aneurysmal evolution may be complicated by aortic rupture [2].

Extension to visceral arteries, complicating 20–30% of AAD, may be responsible for a malperfusion syndrome in downstream organs, including cerebral ischemia.

“Radiographic” malperfusion (evidence of an organ’s vascular compromise) should be distinguished from “clinical” malperfusion or “malperfusion syndrome”, characterized by clinical symptoms, indicating ischemia of downstream organs: the latter carries a much worse prognosis [3].

Reperfusion can cause an increase in intracranial pressure. The brain, very sensitive to ischemia, is susceptible to reperfusion-related edema or hemorrhagic infarction and its complications. The longer the time to reperfusion, the more complications that may occur, thus minimizing the time to reperfusion is therefore essential [3].

Strokes are the major neurological complication, affecting up to one-third of AAD patients, occurring either initially or postoperatively; Zhao et al. reported an incidence of 36.7% of ischemic lesions on diffusion magnetic resonance imaging (MRI) [4]. Stroke is an independent risk factor for early and mid-term mortality [5].

Strokes complicating AAD are of ischemic type, most often multifactorial, due to:−An extension of the dissection to the supra-aortic branches, with carotid or vertebral occlusion, by compression of the true lumen, or thrombo-embolic mechanism arising from the false lumen;−Often coexisting with systemic hypotension, resulting in decreased cerebral perfusion [4,6].

Cerebral malperfusion (CM), to be distinguished from ischemic stroke, complicates 11% of AAD [3,7]. In the study by Geirsson et al., its presence in the preoperative period increased in-hospital mortality from 9.5 to 50%, the risk of postoperative stroke from 4.5 to 46.7%, the occurrence of coma from 4.5 to 40%, and confusion from 29.6 to 73.3% [8].

Extension of dissection to the common carotid arteries (CCA) has an incidence of 30%. It is associated with a significant increase in preoperative neurological deficits (23%, compared to 3% without extension) [9,10]. The right CCA is most frequently affected, in conjunction with the brachiocephalic artery [11].

Appearance of new deficits postoperatively may result from the increase in preoperative malperfusion, intraoperative hypoperfusion, or a perioperative embolic event [12]. A transient deficit may be observed in association with reperfusion edema.

Aortic and carotid surgery is the standard treatment in case of cerebral malperfusion.

Amr et al. noted that isolated hypoperfusion should not be a contraindication for surgery, regardless of its severity [13].

If CM occurs, the aim of surgery is to restore cerebral perfusion to minimize the risk of long-term complications. The usual attitude is to replace the arch and repair the dissected common carotid artery, hoping that repair of the dissection will correct the malperfusion [10].

There is no definitive recommendation regarding the place of carotid stenting (CS) in case of CM complicating an AAD extended to the carotid arteries [14].

Several authors have reported stenting management of the dissected carotid artery before surgery for downstream CM, but data in the literature remain limited.

Carotid angioplasty can be performed more rapidly than surgery, under local anesthesia. The goal is to cover the entire dissected portions with stents, allowing obliteration of the false lumen and re-expansion of the true lumen.

On the other hand, the benefit of CS in the aftermath of aortic surgery (AS) has also been reported, because of residual dissection with true lumen stenosis, resulting in symptomatic cerebral hypoperfusion.

Postoperative malperfusion syndrome is an independent predictor of early mortality according to the multicenter study by Czerny et al. [7,10]. Similarly, asymptomatic residual radiological malperfusion after surgical repair, has been associated with long-term neurological sequelae, with a fourfold relative risk of transient ischemic attack or stroke [3].

Our objective is to report our local experience of AAD patients with CM treated with CS before or after AS and to perform a systematic literature review of AAD patients with CM treated either with CS before or after AS or with aortic and carotid surgery alone (ACS).

## 2. Materials and Methods

### 2.1. Ethics

For the cohort study, as this was a non-interventional retrospective study of routinely acquired data, written informed consent for this study was not necessary. The manuscript was prepared following the Preferred Reporting Items for Systematic-Reviews and Meta-Analyses (PRISMA) guidelines.

### 2.2. Local Center Patients’ Selection

We conducted a retrospective search in our center’s PACS database, from January 2018 to June 2022.

The inclusion criteria were: (1) adult patients, aged 18 years old or more, (2) with uni- or bilateral carotid dissection, following acute type A aortic dissection, without intracranial vessel occlusion, (3) associated with brain hypoperfusion in the territory of the dissected vessels, attested by either hypoperfusion proved with perfusion–CT or perfusion–MRI sequence, or typical junctional stroke pattern on diffusion MRI sequence, (4) patients who underwent CS before or after AS.

Patients with carotid or large intracranial vessel occlusion due to an embolic mechanism, or who were managed conservatively or with thrombolysis were excluded.

### 2.3. Literature Review

A systematic review of the literature was performed in the PubMed database separately by two authors (J-FA and TA). The keyword associations used were “Aortic dissection AND Carotid stenting”, “Aortic dissection AND carotid angioplasty”, “aortic dissection AND carotid plasty”, “Aortic dissection AND stroke”, “Acute disease AND aneurysm, dissecting AND cerebrovascular disorders”. All articles dating from 1990 to the research date (February 2022) were included for screening.

The inclusion and exclusion criteria for selecting the articles were the same.

Titles and abstracts were screened. Studies meeting prespecified inclusion criteria were reviewed in full. PRISMA guidelines were strictly adhered. Two authors completed the quality assessment and evaluation of bias according to specific guidelines.

We evaluated the outcome of these patients in terms of mortality, residual disability, and intra- and post-procedural complications.

Risk of bias was assessed according to the Joanna Briggs Institute Critical Appraisal tools for case reports, and to the Cochrane Collaboration tool for cohort studies (details in online Appendix A).

### 2.4. Data Extraction and Expression of Results

Baseline characteristics were displayed as absolute number (percentage) or mean (SD) or median (interquartile range [IQR], e.g., 25th–75th quantiles).

Extracted data included bibliographic information, type of paper, stated aim, topic/focus of systematic review, study/review methodology, description of reported involvement, baseline characteristics and results about treatment and outcome.

## 3. Results

### 3.1. Local Center Patients’ Selection (n = 9)

In total, nine patients were included in the study: four who underwent CS before AS, and five following AS, all of them because of neurologic deficits, attributed to a hemodynamic brain hypoperfusion. The characteristics of these patients are detailed in Table 1.

The nine patients managed in our center had a median age of 72 years (IQR 67.5–75.5), with a male predominance.

All patients had an mRS score of 0 before dissection. The median ASPECTS score was 9 (IQR 8–10) (n = 3). Following CS, the neurological deficit partially regressed in 66.7% of the patients. An increase in neurological deficit occurred in 11% for various reasons detailed in Table 1. During carotid stenting, three patients presented with distal emboli, e.g., a distal A2 embolus without endovascular rescue, a sylvian M2 embolus treated by thrombectomy; multiple distal emboli were not accessible by thrombectomy.

Following CS, three patients (33%) presented new ischemic lesion: a junctional infarct and a sylvian punctiform spot, distal diffuse ischemic lesions. Of these three patients, one patient presented with clinical worsening. All surviving patients had good stent permeability at follow-up.

### 3.2. Literature Review (n = 73)

A total of 5307 articles were identified. After exclusion based on the abstract, 703 articles were retained for detailed review. Finally, 19 articles matched all the criteria, 17 case reports, and 2 retrospective series were included. None of the studies were prospective; see flowchart of the included studies (Figure 1). The characteristics of included studies are shown in Table 2.

Patients had a median age of 64.5 years (IQR 57–70) for the case reports [11,13,14,15,16,17,18,19,20,21,22,23,24,25,26,27,28], and 60.5 years and 68 years in the cohorts [12,17]; 52% were female. A history of arterial hypertension (AH) was found in 83% of the cases.

All patients had an mRS score of 0 before dissection. The mean initial NIHSS score (all items combined) was 13.6 ± 11.2. The median time from symptom onset to treatment was 6 h (IQR 3.5–15).

**Table 2 jpm-13-00058-t002:** Summary of included studies.

Study	Year	Study Type	Number of Patients Included	First Treatment
Schönholz et al. [15]	2008	Case report	1	Surgery
Chahine et al. [16]	2018	Case report	2	Surgery
Morihara et al. [17]	2016	Case report	1	Surgery
Matsumoto et al. [18]	2016	Case report	2	Surgery
Amr et al. [13]	2016	Case report	2	Surgery
Hong et al. [19]	2005	Case report	1	Surgery
Karawabuki et al. [20]	2006	Case report	1	Surgery
Kim et al. [29]	2006	Case report	1	Surgery
Ueyama et al. [22]	2007	Case report	1	Surgery
Roseborough et al. [11]	2006	Case report	1	Surgery then stenting
Sakaguchi et al. [23]	2005	Case report	1	Surgery
Usui et al. [24]	2021	Case report	1	Surgery
Fukuhara et al. [25]	2021	Case report	1	Surgery
Funakoshi et al. [26]	2020	Case report	2	Stenting then surgery
Heran et al. [27]	2019	Case report	1	Stenting then surgery
Popovic et al. [14]	2016	Case report	1	Stenting then surgery
Casana et al. [28]	2011	Case report	1	Surgery then stenting
Fichadaya et al. [12]	2022	Cohort	10	Surgery
Morimoto et al. [30]	2011	Cohort	41	Surgery

### 3.3. Aortic and Carotid Surgery (ACS) Group (n = 72)

The mean age of the group was 65.5 ± 11.0 years; 53% were female.

The median initial NIHSS score was 16 (IQR 5–19) for the case reports and 8 for the Morimoto et al. cohort [31]. The mean NIHSS for all articles was 11.3 ± 8.5. The median time from symptom onset to treatment for the case reports was 5.5 h (IQR 3–12), and the mean time for all articles was 21 ± 39.3 h.

One patient died during surgery [15] of an uncontrollable hemorrhage complication. No other intraoperative complications were reported.

A new persistent neurological deficit occurred for 28% of the operated patients. A new ischemic lesion on postoperative DWI–MRI occurred for 49% of the patients.

Overall, 13% of the patients (15/53) died during hospitalization (mRS 6). For the other patients, the median mRS score was 3 (IQR 2–3) at discharge, and 0 (IQR 0–0.5) at last follow-up. The mean follow-up time was 53.4 months.

Individual patient data for this group are available in Appendix A.

### 3.4. Carotid Stenting (CS) before Aortic Surgery (AS) Group (n = 8)

The mean age of the group was 65.3 ± 13.1 years; 25% were female. A history of AH was found in 66% of cases. All patients had an mRS score of 0 before dissection.

The median initial NIHSS score was 14 (IQR 11–21). The mean NIHSS was 14.3 ± 8.0. The median time from symptom onset to treatment was 5.5 h (IQR 3–18), with a mean duration of 13.6 ± 17.8 h.

No deaths during CS or surgery occurred. No patient had any intra-CS or intra-operative complication. Partial regression of neurological deficits was achieved in six patients (86%). The median post-CS NIHSS was 4 (n = 3 patients).

A new ischemic lesion on post-CS imaging occurred in 2 patients.

One patient died before aortic surgery, at day 2, of mesenteric ischemia; 1 patient did not undergo surgery because of a history of Bentall surgery. The other patients received aortic replacement afterwards.

None of the operated patients presented new neurological deficits or new ischemic lesions on postoperative imaging.

One patient died postoperatively during hospitalization from mediastinitis on day 35 [16]. For the other patients, the median mRS score was 2 at discharge, and 1 at follow-up. The mean follow-up time was 6.8 ± 3.6 months. All stents were patented at follow-up.

An example of one of our treated patients is shown in Figure 2.

Individual patient data for this group are available in Appendix A.

### 3.5. Carotid Stenting (CS) after Aortic Surgery (AS) Group (n = 7)

The mean age of the group was 68.7 ± 5.8 years; 71% were female. A history of AH was found in 66% of cases.

All patients had an mRS score of 0 before dissection.

The initial NIHSS score of all patients was 0. The median time from symptom onset to surgery was 6 h (IQR 6–8), with a mean duration of 13 ± 17.2 h.

All patients presented a new neurological deficit postoperatively, with six of the seven patients presenting new lesions and a hypoperfused cerebral territory on imaging, justifying management by stenting.

The median surgery to stenting time was 3 h (IQR 1–14). During the procedure, three patients presented distal emboli despite the use of an FilterWire EZ (Boston Scientific, Marlborough, MA, USA) distal protection filter in all cases:−A 78 yo patient presented with a distal A2 embolus, too distant to be accessible by thrombectomy. Three 9 × 30 mm Carotid Wallstents were used, covering the entirety of the brachiocephalic trunk and right common carotid artery. After deployment of the first two Carotid Wallstents, an intra-stent thrombosis occurred, and was immediately and successfully treated by aspiration, and was not recurrent. Antiplatelet treatment was started the next day.−A 74 yo patient presented a sylvian M2 embolus at the end of the procedure, successfully treated by thrombectomy immediately after stenting with three Carotid Wallstent (7 × 40 mm, 9 × 50 mm and 5 × 30 mm) and a Smart Control 14 × 40 mm. Antiplatelet treatment and preventive low molecular weight Heparin were started the next day.−A 62 yo patient presented multiple distal emboli not accessible to thrombectomy. Antiplatelet treatment and curative low molecular weight Heparin were started 12 h post-procedure.

On postoperative imaging, the last two patients presented, respectively, a junctional infarct and distal embolic lesions.

Partial regression of neurological deficits was obtained in four patients (57%).

Only one patient presented a neurological worsening in the post-stenting period, related to reperfusion edema, which rapidly completely resolved.

One patient was still hospitalized at the time of writing, a few days after stenting, and the outcome could not be studied.

Two patients died during hospitalization. In the surviving patients, the median mRS score was 2 at discharge and 1 at follow-up.

Individual patient data for this group are available in Appendix A.

## 4. Discussion

Regarding the patients from our local cohort (n = 9), the mean age was 71.2 ± 5.3 years, with a male predominance, a history of AH was found in 78%. The median ASPECTS score was 9 (IQR 8–10). Following CS, neurological deficits partially regressed in 0.67% (2/3) of the patients and remained stable in 0.33 (1/3) of the cases. During CS, three patients presented procedural complications: multiple distal emboli, an A2 emboli, and a homolateral sylvian M2 emboli, the latter being successfully treated by thrombectomy. The first patient had an immediate intra-stent thrombosis, successfully treated by aspiration. All surviving patients had good stent permeability at follow-up.

Experience in carotid stenting in the context of stroke derives from three situations: thrombectomy in thromboembolic stroke, urgent carotid repair in a traumatic setting, and isolated carotid artery stenosis or dissections (in rare cases).

The meta-analysis of Fabre et al. in the context of primitive carotid dissections, reported a technical success rate of 99.1% in 201 patients, with rare complications, including one embolic stroke, two subarachnoid hemorrhages, a transient vasospasm in two patients.

During follow-up, only 3.3% of patients developed intimal hyperplasia or intrastent stenosis; 2.1% had recurrent TIA in the territory of the stented vessels [32].

In the context of AAD extended to the carotid arteries, a limited number of case reports have been published in the literature, with stenting *before* [14,26,27], *during* [31,33,34], *or after* [11,21,28,34,35,36,37,38] *aortic* surgical repair: in all reported cases, technical success was achieved, with complete exclusion of false lumen, and stent patency at follow-up. All patients showed improvement or complete resolution of neurological symptoms. Only one case of death is reported, unrelated, from mediastinitis.

Recently, Mukherjee et al. reported three patients treated by retrograde carotid stenting after aortic repair, with residual stenosing carotid or brachial artery dissections. Venous stents (Boston Scientific VICI) were chosen because of their greater radial strength than the stents normally used, with the goal of a better obliteration of the false lumen [34].

Carotid angioplasty can be performed more rapidly than surgery, under local anesthesia.

Funakoshi et al. summarized the various challenges of performing carotid angioplasty-stenting preoperatively [26]:

(1) The need for double antiplatelet therapy in the immediate post-procedure period, which may lead to postpone the urgent aortic replacement surgery, which can only be carried out under mono-antiaggregation. (2) A risk of post-procedure reperfusion edema.

A prior perfusion imaging (CT or MRI) allows to evaluate this risk. (3) The technical difficulty of catheterizing the true lumen. (4) Restriction of future surgical options: when stents cover the origin of aortic branches total arch replacement is impossible. Both patients in the article were treated with hemi-arch replacement for this reason. (5) The risk of per-procedure distal emboli. However, according to a meta-analysis by Fabre et al., the risk of distal emboli is low for the treatment of dissections: for patients who benefited from a stenting treatment of primary internal carotid dissections, only 1 out of 201 treated patients presented an embolic stroke [32].

After the procedure, most patients are put on double antiplatelet therapy for 1–6 months, then on permanent single antiaggregant [32].

A summary of the benefits and drawbacks of the procedure is presented in Table 3.

The study by Fukuhara et al. found a significant difference in morbidity and mortality with ICA vs. CCA occlusion: every one of the ICA patients developed rapid cerebral edema with herniation, and died during hospitalization, whereas 79% of patients with occlusion limited to CCAs (uni- or bilateral) survived, and only one patient developed cerebral edema. These results suggest that occlusion of an ICA may be a marker at risk for cerebral edema and herniation [25].

The residual dissection of postoperative supra-aortic branches is associated with long-term neurological complications [11]. The prospective study by Neri et al., which followed 42 of these patients for a median of 3.17 years, found an incidence of neurological events of 30.9%, including stroke in 18 patients, with a relative risk of 3.99, all of which occurred in the territory of the initial dissected artery [39].

To the best of our knowledge, our study is the first aiming to synthesize the literature on carotid stenting for hemodynamic injury in the context of AAD.

Our study has many limitations. The included articles were mostly case reports, a source of important biases, notably publication bias. The patients’ selection from our center was retrospective, non-randomized, and monocentric. Due to the small number of patients included, statistics comparing the different groups could not be performed. All the articles studied did not evaluate the long-term outcome of patients and did not allow us to obtain any hindsight on mortality or recurrence of symptoms in the long term.

## 5. Conclusions

Preoperative carotid stenting for hemodynamic CM by true lumen compression appears to be feasible, and could be effective and safe, although there is still a lack of evidence, due to the absence of comparative statistical analysis. The literature, albeit growing, is still limited, and prospective comparative studies are needed.

## Figures and Tables

**Figure 1 jpm-13-00058-f001:**
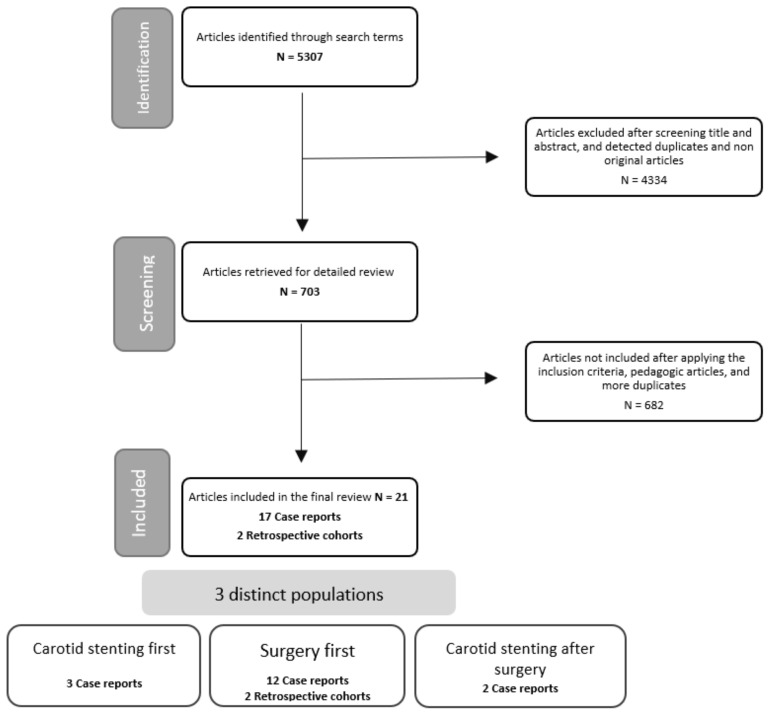
Flowchart of included studies.

**Figure 2 jpm-13-00058-f002:**
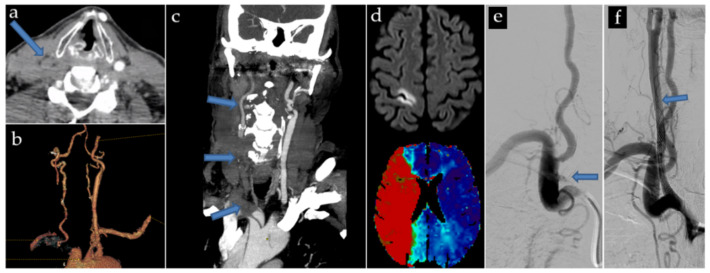
Local cohort case of a 72-year-old, who suffered a brutal left hemicorporal deficit. CT scan showed an AAD extended to the LCCA, with a subocclusive compression of the true lumen (**a**–**c**). Initial MRI showed a significant mismatch between the limited cytotoxic lesion of the rolandic area, and the hypoperfusion of the entire sylvian territory (**d**). Preoperative angioplasty was decided (**e**) The initial occlusion is shown. Stenting allowed to re-expand the true lumen, with coverage of the entire carotid dissection (**f**). The patient underwent aortic surgery 6 h later, without complications, and recovered gradually after. Post-surgical MRI (not shown) found no new cytotoxic lesion, and showed complete resolution of the hypoperfusion.

**Table 1 jpm-13-00058-t001:** Local patients characteristics.

Patient	1	2	3	4	5	6	7	8	9
Treatment	Carotid stenting before aortic surgery				Carotid stenting after aortic surgery				
Age	71	71	72	72	78	64	71	74	62
Sex	M	M	M	M	F	M	F	F	F
Arterial hypertension	yes	No	yes	yes	yes	yes	yes	no	yes
Current smoking	No	No	Yes	No	No	Yes	No	No	yes
Pre-stroke MRS	0	0	0	0	0	0	0	0	0
NIHSS on presentation	21	NA	10	3	NA	0	0	0	0
ASPECTS score	7	10	8	9	10	10	8	NA	NA
Supra-aortic branch involved	RCCAand ICA, LCCA and ICA	RCCA and ICA	RCCA and ICA	RCCA	RCCA	LCCA	RCCA	RCCA and ICA, LCCA	RCCA
Per-procedure complications	No	No	No	No	Yes (common carotid intra-stent thrombosis treated by aspiration, A2 distal emboli)	No	No	Yes M2 (sylvian emboli)	Yes (distal emboli)
New lesion on post-procedure MRI	No	No	No	Yes	No	No	No	Yes (junctionnal infarct)	Yes (diffuse distal ischemic lesions)
New neurological deficit post-procedure	No	No	No	No	No	Yes (transient majoration due to reperfusion oedemea)	No	No	No
Post-procedure neurologic deficit regression	Partial	Partial	Partial	No	No	Partial	Partial	No	No
Survived	Yes	No Mesenteric ischemia (day 2)	Yes	Yes	No Neurologic degradation (day 8)	Yes	Yes	No Cardiac arrest (day 4)	NA
MRS at discharge	2	-	4	NA	-	1	4	-	NA
MRS after 90 days	1	-	3	1	-	1	1	-	NA
Sent permeability at follow-up	Yes	Yes	Yes	Yes	Yes	Yes	Yes	Yes	NA

**Table 3 jpm-13-00058-t003:** Comparison of benefits and drawbacks.

Carotid Stenting First	Aortic and Carotid Surgery First
Benefits	
Local anesthesia	Faster repair of ascending aorta
Faster installation (15 min of installation on table)	Correction of systemic hypotension
Treatment of distal internal carotid artery dissections	
Complete exclusion of the false lumen	
Drawbacks	
Double antiaggregation post-procedure: postponed surgery	Longer preparation
Difficulty in catheterizing the true lumen	General anesthesia
Restriction of surgical options: impossibility of total arch replacement when stents cover the origin of supra-aortic branches	Risk of intraoperative hypoperfusion under cardiopulmonary bypass
Risk of distal embolization per procedure	Risk of residual dissection and/or cerebral malperfusion if distal repair is impossible

## Data Availability

The data presented in this study are available on request from the corresponding author.

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
