# Peer review of "Stenting in Brain Hemodynamic Injury of Carotid Origin Caused by Type A Aortic Dissection: Local Experience and Systematic Literature Review"

_jpm, 2022, doi:10.3390/jpm13010058_

Round 1

Reviewer 1 Report (Previous Reviewer 1)

The authors responses have released my concerns, I have no more questions.

Author Response

Thank you very much for your review.

For this round, we corrected four missing explanations for abbreviations (CS, CM, AS, MRI) according to the remarks of Reviewer n°2.

Reviewer 2 Report (Previous Reviewer 2)

There is still no explanation of CS, CM , AS, MRI  in the first parts of edited manuscript.

There are no new comments regarding this manuscript

Author Response

Thank you very much for your review. We added the 4 explanations for the abreviations (CS, CM, AS, MRI), and they are now correctly explained in the new version.

This manuscript is a resubmission of an earlier submission. The following is a list of the peer review reports and author responses from that submission.

Round 1

Reviewer 1 Report

This manuscript performed a literature review of AAD patients with CM treated either with CS before or after AS or with aortic and carotid surgery alone (ACS). The authors conducted a meta-analysis-like method to review published articles, while did not provide any useful or novel conclusions. In addition, many places in their study could also be improved.

1.     It is obviously not enough for authors to search articles only from PUBMED database. Since there are limited article numbers on this topic, the author should also investigate other databases such as Embase, Cochrane Library, OVID, etc.

2.     The “Introduction” needs to be improved, as too many redundant and unrelated content in there, such as the pathogenesis of AD and the pathophysiology of AD related stroke.

3.     Line 154, the reference was incorrectly cited.

4.     “Figure-1” showed that in their study there were two case reports about conservative treatment, but why there was only one in “Table-2”? 

5.     Any evaluation of the quality, bias of the included literature?

6.     It is recommended to add patients who only receive ACS surgery in the selection of local center patients, and conduct comparative analysis between groups.

7.     CS is a useful treatment for AD related cerebral malperfusion. However, endovascular treatment for acute AD is a high-risk operation, and the indications and methods are still controversial. Therefore, the indications of the surgery need to be provided.

8.     The authors reported 9 patients in local center who received CS for AD related cerebral malperfusion, and 3 of them suffered distal emboli after operation. Since distal stent blockage is one of the key factors in endovascular treatment, the detailed description of the three patients is needed, including the blockage time, anticoagulation treatment, and stent size, length and brand.

9.     Line 372-373, this conclusion is obviously inappropriate without statistical comparative analysis.
